# Advancements in Locoregional Therapies for Unresectable Intrahepatic Cholangiocarcinoma

**DOI:** 10.3390/curroncol32020082

**Published:** 2025-01-31

**Authors:** Conor D. J. O’Donnell, Umair Majeed, Michael S. Rutenberg, Kristopher P. Croome, Katherine E. Poruk, Beau Toskich, Zhaohui Jin

**Affiliations:** 1Department of Medicine, Division of Hematology-Oncology, Mayo Clinic Florida, Jacksonville, FL 32224, USA; 2Department of Radiation Oncology, Mayo Clinic Florida, Jacksonville, FL 32224, USA; 3Department of Transplantation, Mayo Clinic Florida, Jacksonville, FL 32224, USA; 4Department of Surgical Oncology, Mayo Clinic Florida, Jacksonville, FL 32224, USA; 5Department of Interventional Radiology, Mayo Clinic Florida, Jacksonville, FL 32224, USA; 6Division of Medical Oncology, Mayo Clinic College of Medicine, Rochester, MN 55905, USA

**Keywords:** intrahepatic cholangiocarcinoma, selective internal radiation therapy, stereotactic body radiation therapy, proton beam therapy, liver transplantation, hepatic artery infusion pump

## Abstract

Intrahepatic cholangiocarcinoma is an aggressive malignancy with rising incidence and poor outcomes. This review examines recent advancements in locoregional therapies for unresectable intrahepatic cholangiocarcinoma, focusing on external beam radiotherapy, transarterial radioembolization (TARE), hepatic artery infusion pump (HAIP) chemotherapy, and liver transplantation. Stereotactic body radiation therapy and proton beam therapy have shown promise in achieving local control and improving survival. TARE, with personalized dosimetry, has demonstrated encouraging results in select patient populations. HAIP chemotherapy, primarily studied using floxuridine, has yielded impressive survival outcomes in phase II trials. Liver transplantation, once contraindicated, is now being reconsidered for carefully selected patients with localized disease. While these locoregional approaches show potential, randomized controlled trials comparing them to standard systemic therapy are lacking. Patient selection remains crucial, with factors such as liver function, tumor burden, and molecular profile influencing treatment decisions. Ongoing research aims to optimize treatment sequencing, explore combination strategies with systemic therapies, and refine phenotype identification and patient selection criteria. As the landscape of intrahepatic cholangiocarcinoma management evolves, a multidisciplinary approach is essential to tailor treatment strategies and improve outcomes for patients with this challenging disease.

## 1. Introduction

Intrahepatic cholangiocarcinoma is the second most common cause of liver cancer, accounting for up to 15% of cases worldwide [1]. The incidence rate was 123,000 cases per year globally in 2018 [2]. Incidence and mortality rates have been increasing in Western countries in recent decades [3,4]. As with several other gastrointestinal cancers, there is a concerning trend of increases in early-onset disease in young adults [5]. Intrahepatic cholangiocarcinoma accounts for 20–25% of all cases of cholangiocarcinoma and is anatomically defined by its origin in second-order or higher bile ducts. There are distinct differences in the molecular profile compared to extrahepatic cholangiocarcinoma, such as a higher incidence of IDH1 mutations and FGFR2 fusions.

Upfront surgical resection remains the treatment of choice for intrahepatic cholangiocarcinoma. However, only 12–17% of patients with intrahepatic cholangiocarcinoma undergo surgical resection [6,7]. This may be explained by the insidious growth of non-Klatskin tumors before symptoms of biliary obstruction arise. The majority of patients with intrahepatic cholangiocarcinoma present with locally advanced disease rather than distant metastatic disease [8,9]. For those that are able to undergo surgical resection, recurrence rates are high, with 5-year recurrence-free survival between 9 and 34% [10,11]. Adjuvant therapy with single-agent capecitabine is recommended in guidelines due to a survival advantage in the per-protocol analysis of the BILCAP trial [12]. Median overall survival (OS) in the capecitabine arm was 52.3 months versus 36.1 months with observation (HR 0.79, 0.63–1.00), though the primary endpoint was not met in the intention-to-treat population. Most benefits appear to come from the delaying of recurrence rather than a reduction in recurrence. For intrahepatic cholangiocarcinoma specifically, a recent large retrospective analysis failed to show improvement in survival from adjuvant capecitabine [13]. Adjuvant trials of gemcitabine [14,15] or gemcitabine–oxaliplatin [16] for patients with cholangiocarcinoma were also notably negative. The multicenter international phase III ACTICCA-1 trial of gemcitabine–cisplatin in this setting has completed accrual, with results anticipated in the near future [17].

One major challenge with surgical resection is the difficulty of achieving clear margins. Around 20% of patients will have a microscopically positive margin (R1 resection), and another large proportion (20%) will have a margin < 5 mm [18]. The survival benefit from surgery with an R1 resection is unclear [19] with few long-term survivors: 3-year survival at 22% and 5-year survival at 13.1% in one large series [18,20]. There is incremental worsening of disease-free survival (DFS) and OS as the margin width decreases [18]. An additional challenge, given the typical locally advanced presentation of intrahepatic cholangiocarcinoma, is the requirement of adequate future liver remnant in patients after resection to prevent post-hepatectomy liver failure [21]. Even with R0 resections with an adequate liver remnant, poor tumor biology remains a competing risk for early disease recurrence. Patterns of recurrence after resection do suggest that the liver is the most common site of recurrence for intrahepatic cholangiocarcinoma: 61.5% of recurrences are limited to the liver, whereas 38.5% recur with a component of extrahepatic disease [18]. In patients with primary sclerosing cholangitis (PSC), liver resection for intrahepatic cholangiocarcinoma may also be limited by the amount of underlying liver disease secondary to PSC [22].

For those with disease that is deemed unresectable, systemic therapy has been considered the cornerstone of treatment. Historical data suggest that median survival in this setting without systemic therapy is around 4–5 months [23,24]. Overall outcomes with systemic therapy still remain poor. Gemcitabine and cisplatin as a front-line systemic therapy has a median OS of less than 1 year in patients with metastatic or unresectable biliary tract cancers [25]. A post hoc analysis of the ABC trials restricted to the subgroup of patients with unresectable locally advanced intrahepatic cholangiocarcinoma included 34 patients and found a median OS of 16.7 months with a 3-year OS rate of 2.8% [26]. Other publications show similarly low rates of long-term disease control with systemic therapy alone in this setting [7]. Furthermore, we know that roughly 40% of patients treated with this gemcitabine–cisplatin treatment in clinical practice do not fulfill the inclusion criteria outlined in the ABC trials, primarily due to hepatic derangement [27]. Limited data exist on the effectiveness of neoadjuvant chemotherapy in downstaging patients to resectability in this setting; however, those who convert to resectable disease after neoadjuvant therapy seem to have similar [28] or improved [29] survival compared to those who undergo upfront surgery.

Recently, immune-checkpoint inhibition with durvalumab [30] or pembrolizumab [31] added to gemcitabine–cisplatin showed a modest improvement in overall survival by 1.3 and 1.8 months, respectively. There was, however, a doubling of 3-year survival in the TOPAZ-1 trial, with 14.6% of patients alive in the durvalumab arm versus 6.9% with chemotherapy alone [32]. Other escalation strategies in triplet chemotherapy, such as FOLFIRINOX [33] and gemcitabine–cisplatin–nab-paclitaxel [34], have failed to show benefits.

Targeted therapies have emerged as treatment options for select patient populations. Around 20% of intrahepatic cholangiocarcinoma tumors carry mutations in IDH1 and may benefit from ivosidenib with a median OS of 10.3 months in a later line setting and 3–5-months of improvement over a placebo [35,36]. FGFR2 fusions occur in about 14% (range 10–20%) of cases [37] with several therapies approved or in development [38]. For instance, futibatinib has shown an overall response rate (ORR) of 42%, a duration of response of 9.7 months, and a median OS of 21.7 months [39]. BRAF mutations occur in about 4% of patients [40], with dabrafenib–trametinib showing an ORR of 47%, PFS of 9 months, and OS of 14 months in patients with the BRAF V600E mutation [41]. HER2 amplification is less common with intrahepatic cholangiocarcinoma (5% versus 20% for extrahepatic cholangiocarcinoma) [42] with several therapies such as trastuzumab deruxtecan [43] (ORR 36.4%), zanidatamab (ORR 41%) [44] and pertuzumab–trastuzumab (ORR 23%) [45] among the agents with demonstrated activity [46]. Approximately 2% of patients with cholangiocarcinoma will be mismatch-repair deficient [47] and may more profoundly benefit from immunotherapy [48]. Despite these advances, virtually all patients will progress on systemic therapy, and for patients with intrahepatic cholangiocarcinoma, over 70% will die secondary to progressive disease in the liver and liver failure [49].

These facts provide a strong rationale for using locally directed therapies in intrahepatic cholangiocarcinoma, with evidence suggesting that the ability to control disease within the liver improves survival [50]. Yet, major guidelines currently diverge with respect to their recommendations for locoregional therapies in this setting. For example, the European Association for the Study of the Liver (EASL) guidelines list intra-arterial therapies as a reasonable option for patients with unresectable liver-limited disease and liver transplantation for select cases of a single lesion ≤2 cm [51]. The National Comprehensive Cancer Network (NCCN) guidelines list chemoradiation and arterial-directed local therapy as front-line options for locally advanced unresectable disease [52]. The European Society for Medical Oncology (ESMO) guidelines place local therapies only after standard systemic treatment for these patients [53]. The American Association for the Study of Liver Disease (AASLD) guidelines state that data are insufficient to recommend locoregional therapy as a standard option for locally advanced, unresectable intrahepatic cholangiocarcinoma and suggest liver transplantation should only be considered under a research protocol [54].

This article will review the recent data for external radiation, transarterial radioembolization (TARE), hepatic artery infusional chemotherapy, and liver transplantation for unresectable intrahepatic cholangiocarcinoma. We will highlight the limitations of the currently available studies that lead to such discrepancies in recommendations and attempt to identify populations and disease presentations most likely to benefit from specific modalities of locoregional therapy. Definitions of resectability and advances in surgical techniques redefining these definitions, transarterial chemoembolization, and non-radiation-based forms of ablation are beyond the scope of this review.

### 1.1. External Beam Radiotherapy

An evolving list of non-invasive forms of radiation have been used for the treatment of unresectable intrahepatic cholangiocarcinoma [55]. The clinical benefit of external beam radiation with conventional dosing for intrahepatic cholangiocarcinoma has been suggested from registry data [56]. Compared with no treatment, treatment with radiation alone was associated with an improved median survival and was associated with a 31% reduction in the risk of death. This was, in fact, comparable to the risk reduction for death seen with surgery alone (38%) in this analysis.

A single phase II randomized control trial (FFCD 99-02) [57] compared front-line chemotherapy with gemcitabine–oxaliplatin to chemoradation (50 Gy in 25 fractions with infusional 5-fluorouracil and cisplatin) in locally advanced cholangiocarcinoma. Approximately half the included patients had intrahepatic cholangiocarcinoma. The trial closed in 2010 due to slow accrual (34 of the planned 72 patients accrued). Both PFS and OS were numerically worse in the chemoradiation arm, though this finding was not statistically significant. It was concluded that the 50 Gy dose level was not strong enough to cure tumors, and concern of radiation-induced hepatitis limited the applicability of conventional dosing for larger tumors.

Stereotactic body radiation therapy (SBRT) is characterized by the delivery of highly conformal radiation and a hypofractionated course, allowing for higher doses of radiation to be more precisely delivered. A single-institution retrospective analysis of radiation treatment in 79 patients with inoperable cholangiocarcinoma highlights the importance of dosing [58]. In this series, most patients (89%) had received prior systemic therapy, and the median size of the primary tumor was 7.9 cm. A threshold biological equivalent dose (BED) of 80.5 Gy was established for ablative dosing. When this dosing was achieved, the 3-year OS rate was an impressive 73% (which is higher than many series on surgical resection) versus 38% for cases not reaching the 80.5 Gy threshold. Local control rates were similarly improved to 78% vs. 45% (*p* = 0.04) above and below this threshold. Treatment was shown to be very safe, with no cases of radiation-induced liver disease documented. Interestingly, the baseline primary tumor size, the presence of satellite tumors, regional nodal or extrahepatic metastases were not found to be prognostic, while radiation dose was the single most important predictive factor of survival. These findings are again consistent with the fact that progressive disease leading to liver failure is the most common cause of death in this patient population. Other series have also shown improved local control and survival with increased BED [59,60]. Importantly, with improvements in radiotherapy delivery techniques (including SBRT), higher doses with increased BED can be delivered to tumor targets while respecting the dose limits of the liver. This was a limiting factor in historic treatments using radiotherapy for intrahepatic cholangiocarcinoma. The American Society for Radiation Oncology (ASTRO) clinical practice guidelines suggest dose constraints to uninvolved areas of the liver and bowel structures to maintain safety [61]. For example, a mean dose < 15–18 Gy is recommended for the uninvolved liver areas in noncirrhotic patients receiving five fraction treatments compared to <13–15 Gy for those with Child–Pugh A cirrhosis. Maintaining these dose constraints limits the risk of radiation-induced liver disease. Dosing for the stomach and duodenum should be kept below 32 Gy in a five-fraction delivery to minimize the risk of luminal organ ulceration [61].

A 2019 systematic review included 10 studies (1 phase 1 study; 9 retrospective studies) on SBRT for cholangiocarcinoma, including intrahepatic cholangiocarcinoma cases [62]. The median computed BED ranged between 57.6 and 180 Gy, and settings for treatment were quite varied. The 1-year OS rate was 57.1% for studies only including patients with intrahepatic cholangiocarcinoma. The pooled 1-year local control rate was 83.4%.

The first randomized data assessing modern radiation techniques in intrahepatic cholangiocarcinoma have recently been presented at the American Society of Clinical Oncology (ASCO) annual meeting in 2024. The ABC-07 trial [63] is a multicenter randomized phase II trial for patients with inoperable, locally advanced cholangiocarcinoma without progression after 3 months of gemcitabine–cisplatin chemotherapy [63]. Patients were randomized 2:1 to either SBRT (50 Gy in 5 fractions or 67.5 Gy in 15 fractions, based on tumor size) (n = 45) following the sixth cycle of chemotherapy or to complete the standard eight cycles of chemotherapy (n = 24). There was no difference in the primary endpoint, with a median PFS of 8.6 months in the SBRT arm and 9.0 months in the chemotherapy alone arm (HR 1.0). OS was 19.4 months for SBRT and 14.2 months for chemotherapy alone (HR 0.79, *p* = 0.47) at 20.7 months of follow-up. Local control was improved with radiation; however, over 50% of patients developed metastatic progression, and biliary sepsis rates were, in fact, higher in the radiation group. The negative results for the primary endpoint in this trial were disappointing, particularly considering that the inclusion criteria were appropriately restrictive to attempt the selection of patients most likely to benefit from the addition of local therapy, avoiding a common pitfall of other failed trials of radiation therapy in gastrointestinal oncology [64]. However, the majority (85%) of the patients enrolled in this trial had extrahepatic cholangiocarcinoma. We also await the more mature follow-up of OS data to determine if improved local control will lead to reduced death from hepatic failure and overall mortality in the experimental arm.

Proton beam therapy has also been used to treat intrahepatic cholangiocarcinoma. It has distinct physical advantages over standard photon-based radiation, with the deposition of energy at a pre-specified depth without an exit dose. Proton beam therapy can enable high-dose delivery to liver malignancies while maintaining a low dose to surrounding normal tissues, including the uninvolved liver, bowel, and adjacent heart (for lesions in the liver dome). A prospective single-arm study evaluated the efficacy of this treatment modality [65]. In total, 39 patients with unresectable or locally recurrent intrahepatic cholangiocarcinoma (with a maximum tumor dimension up to 12 cm for a solitary tumor, 10 cm for two tumors, and 6 cm for three tumors) were treated with a dose between 58.05 and 67.5 Gy-equivalent delivered in 15 fractions. In total, 87.2% of the included patients had a single lesion, and only 10.3% of patients had cirrhosis. Treatment was safe, with the rate of grade-3 radiation-related toxicities at 7.7%, and only 3.6% had worsening Child–Pugh scores following treatment. The local failure rate was 15% (6/39). The median PFS was 8.4 months, and the 2-year PFS rate was 25.7%. The median OS was 22.5 months, with a 2-year OS rate of 46.5%. A recent retrospective analysis in a more heterogenous cohort of patients with cholangiocarcinoma showed similar findings with a local failure rate of 12% and median OS of 19.3 months [66].

Another phase III trial with a similar design to ABC-07, the NRG GI-001 trial (NCT0220042), compared a 15-fraction radiation schedule (with photons or protons) versus observation following 6 months of gemcitabine–cisplatin chemotherapy in unresectable localized intrahepatic cholangiocarcinoma terminated early due to a lack of accrual. The question of protons versus photons for liver cancers may be answered in the future by the NRG-GI003 trial (NCT03186898). With further expected advances in precision external radiation, such as carbon ion therapy [67,68], on there way to North American clinics, this will remain an area of active research.

### 1.2. Transarterial Radioembolization

TARE is a procedure by which radiolabeled microspheres, commonly Y-ttrium-90 (Y90) glass microspheres, are administered via the hepatic arteries, delivering radiotherapy directly into the tumor-feeding vasculature. Alternative therapeutic particles such as Y90-resin and Ho-labeled poly(l-lactic acid) microspheres have also been studied [69]. Microsphere injections are preceded by mapping angiography, which entails diagnostic liver angiography with contrast-enhanced cone beam computed tomography (CT) analyses of hepatic arterial perfused volumes or angiosomes, and the intra-arterial injection of technetium-99 m (^99^mTc) macro-aggregated albumin (MAA) as particle simulation, usually 8–15 days prior to TARE treatment. The primary purpose of this mapping procedure has historically been to rule out unacceptable extrahepatic sphere deposition. For example, the non-target delivery of microspheres to the gastrointestinal tract can lead to injury to the gastric mucosa and radiation-induced ulceration and bleeding [70]. As TARE has evolved, however, mapping procedures can now allow for a thorough anatomic assessment of tumor arterial supply. Combining personalized dosimetry based on the MAA sphere distribution as well as ablative dosimetry in expendable volumes of liver has increased safety, improved efficacy and rates of complete pathological necrosis [71]. In hepatocellular carcinoma, this personalized dosimetry approach was prospectively validated in a randomized trial in 2021 [72].

Early experiences with TARE for intrahepatic cholangiocarcinoma have shown variable results. A systematic review found objective response rates ranging from 0% to 36% and median overall survival ranging from 8.7 to 32.3 months [73]. The heterogeneity of these results (largely generated from single-institutional retrospective studies) likely stems from several factors: (1) variabilities in prior therapy; (2) generally high rates of inclusion of extrahepatic disease; (3) TARE protocols from pre- and post-personalized dosimetry eras; (4) variabilities in concurrent systemic therapy administration; (5) inter-operator variability and experience [74,75,76,77,78].

There was a notable prospective single-institution trial of TARE with non-personalized dosimetry used as the sole treatment for unresectable intrahepatic cholangiocarcinoma, which reported an ORR of 71% [79]. Patients with solitary tumors had significantly longer OS (25.9 months versus 10.7 months for multifocal disease (*p* = 0.02). In total, 8% of patients experienced grade 3 toxicities.

The Radiation-Emitting SIR-Spheres in Non-Resectable (RESiN) liver tumors registry is a prospective observational database including 27 centers administering Y90 TARE [80]. Their publication serves as a representative cohort treated between 2015 and 2020 before the validation and adoption of personalized dosimetry. This cohort of 89 patients included 27% with extrahepatic disease and 74% who had received prior systemic therapy. The median OS for the entire cohort after TARE was 14.0 months (12.1–22.3). The OS at 6, 12, and 24 months was 80%, 63%, and 34%. The median PFS was 5.8 months (4.6–7.2) with 6-, 12-, and 24-month PFS at 46%, 27% and 15%. The ORR was 34% (34/70). There were low rates of grade 3–4 toxicities, with <11% of patients experiencing elevated bilirubin, low albumin, or AST/ALT elevation. Two patients developed grade 3 abdominal pain and one developed cholecystitis. Notably, no hepatic abscesses or other infections were reported, even in patients with prior biliary intervention [75,76,77,78]. When considering the degree of pretreatment in this, and other TARE cohort studies, there is likely a group of patients that is naturally selected for TARE by intrahepatic progression on systemic therapy. The systemic therapy options available in the second line for those without targetable alterations are quite poor. FOLFOX [81] has a median OS of 6.2 months and less than a month’s additional benefit over best supportive care. There are conflicting results on the role of nanoliposomal irinotecan and 5-fluorouracil [82,83]. In this context, TARE should be seen as a reasonable option for patients with liver-dominant disease after systemic therapy or for those not fit for systemic therapy.

The “Yttrium-90 Microspheres in Cholangiocarcinoma” (MISPHEC) single-arm phase 2 trial was conducted at seven centers in France and provided the most substantial prospective data for Y90 in a defined patient population [84]. The trial included 41 patients. Notable inclusion criteria were unresectable intrahepatic cholangiocarcinoma with a measurable lesion of at least 2 cm, patients who were noncirrhotic or cirrhotic with Child–Pugh scores less than B8, and an Eastern Cooperative Group performance status of 0 or 1. Patients were allowed to have hilar lymph nodes ≤3 cm or <5 lung nodules (all ≤10 mm), and bilirubin <3 times the upper limit of normal and albumin of at least 2.8 mg/dL. Previous chemotherapy was an exclusion criterion. Four out of forty-five patients were excluded due to extrahepatic fixation on mapping scintigraphy. Chemotherapy with gemcitabine and cisplatin was administered (with gemcitabine reduced to 300 mg/m^2^ during TARE), and the TARE procedure was carried out during the first cycle of systemic treatment and the third cycle in the case of bilobar disease. Personalized dosimetry with the aim of providing at least 205 Gy to the tumor was permitted (the median dose to the tumor was 317 Gy). Baseline characteristics showed that 29% of patients had cirrhosis, the CA 19 9 median was 52 IU, 34% had unifocal disease, 34% had bilobar disease, and 42% had signs of limited extrahepatic disease. The size of treated lesions was not reported.

The trial was positive, meeting its primary endpoint of ORR > 22% by RECIST 1.1 criteria at 3 months, with a 39% ORR, and showed a 3-month disease control rate of 98%, a median PFS of 14 months (8–17), and a median OS of 22 months (14–52).

There were significant toxicities in this trial. In total, 75% of the patients with cirrhosis (9/12) and 17% (5/29) of the patients without cirrhosis experienced hepatic failure (ascites or jaundice). In five of the cases of those with cirrhosis, this was irreversible and occurred in the setting of whole-liver TARE. This led the authors to recommend avoiding the concomitant use of chemotherapy and TARE in patients with cirrhosis. In all, 71% of patients experienced grade 3–4 toxicities, but these were otherwise driven by cytopenias, which were likely attributable to chemotherapy. Despite these toxicities, quality-of-life scores were well maintained overall during treatment [85].

There currently exists a lack of randomized data evaluating TARE for cholangiocarcinoma versus other therapies. The SIRCCA trial (NCT02807181) is a phase 3 trial comparing gemcitabine–cisplatin to TARE alone (without concomitant chemotherapy) in unresectable cholangiocarcinoma. The trial was unfortunately stopped prematurely and could lack the power to provide a definitive answer on the benefit of TARE as a first-line treatment. A recent analysis looked to compare the results seen in the MISPHEC trial to those that may be expected from systemic therapy alone in a similar setting [86]. Individual patient data from 43 patients with liver-limited intrahepatic cholangiocarcinoma that received gemcitabine–cisplatin or gemcitabine–oxaliplatin were identified from the ABC-01/02 [25], ABC-03 [87], BINGO [88], and PRODIGE 38 AMEBICA [33] trials. Propensity scoring was used to limit bias from non-randomized group comparisons between these patients and the MISPHEC cohort [86].

In adjusted OS analyses, the median OS was 21.7 months in the TARE plus chemotherapy group versus 15.9 months in the chemotherapy alone group (HR 0.59, *p* = 0.049). OS at 12 and 24 months were 77% and 41% with TARE plus chemotherapy versus 59% and 32% with chemotherapy alone, respectively. In adjusted PFS analyses, the median PFS was 14.3 months for patients treated with TARE plus chemotherapy versus 8.4 months for patients treated with chemotherapy alone. PFS at 12 and 24 months were 54% and 38% with TARE plus chemotherapy versus 37% and 14% with chemotherapy alone, respectively.

Additional retrospective data using ablative dosimetry has shown promising results. A Mayo Clinic Florida series involved 28 patients with localized unresectable intrahepatic cholangiocarcinoma that received a total of 37 radioembolizations with an ablative dosimetry approach [89]. The median dominant tumor size was 7.3 cm, with 39.3% having bilobar disease. Complete response (CR) was identified in 15 patients (44.1%) and partial response (PR) in 17 (50%) patients using mRECIST. Six patients with unilobar disease were downstaged to resection. Overall, PFS was 8.8 months. The 3-year OS rate was 59%. Multifocal tumors, periductal infiltrating, and intraductal morphology increased tumor size, and poor macrovascular conduit (tumor hypoenhancement) and microvascular conduit were associated with worse outcomes. Experience has suggested that vascular quality for TARE can be affected by the administration of systemic therapies, and thus, upfront TARE may be preferred from a technical standpoint.

Other groups have shown similar results using ablative dosing [90]. The NCCN currently recommends a tumor dose of >205 Gy or ablative dosimetry for TARE when feasible [52].

The above experiences of successful outcomes have led to an interest in using TARE in a neoadjuvant setting as a method to increase the chance of R0 resection in higher-risk cases or as a form of downstaging for unresectable disease. Edeline et al. found that only 4% of patients with liver-only intrahepatic cholangiocarcinoma underwent resection following systemic therapy in prospective clinical trials. In the MISPHEC trial, 22% in the TARE group went on to resection, but this difference was not statistically significant compared to systemic therapy and decreased with propensity score matching [86]. Outcomes appeared to be similar in terms of post-operative complications and overall survival between those with initially unresectable diseases downstaged to resection after neoadjuvant treatment and those that underwent upfront resection. On multivariate analysis from a single-center French study, TARE as the downstaging treatment was associated with a significant post-resection survival benefit [91], whereas neoadjuvant chemotherapy alone was not. Surgery may be more challenging following TARE in some cases, but postulated advantages include the easier delineation of tumor margins, reduced blood loss and less tumoral cell-spreading during manipulation in the setting of induced avascular necrosis [92,93]. A parallel benefit to transarterial radioembolization is the ability to provide a radiation dose to the hepatic future resection site (FRS) in addition to the targeted tumor, resulting in a controlled atrophy of FRS with the concurrent hypertrophy of future liver remnants (FLRs), typically over 3–6 months. This concept, referred to as “radiation lobectomy”, offers the ability to treat tumors with high rates of response, remodel the liver to enable surgical resection, and provide a biological test of time during the liver hypertrophy period for high-risk patients with concurrent systemic therapy. A retrospective study of patients with initially unresectable liver malignancies, including 23% with intrahepatic cholangiocarcinoma, who underwent neoadjuvant radiation lobectomy showed a post-hepatectomy liver failure rate of 3%, an R0 rate of 96%, and median survival of 37.6 months after treatment [21]. Extensive or complete pathologic necrosis was identified in 76% of surgical specimens.

The SIROCHO trial [94] is an ongoing randomized trial of patients with untreated resectable intrahepatic cholangiocarcinoma at high risk for close margins due to (1) margins predicted to be < 1 cm, (2) tumors > 5 cm, or (3) multifocal resectable disease. Patients are randomized to standard-of-care upfront surgery versus neoadjuvant TARE with Y90 glass microspheres and four cycles of capecitabine. The primary endpoint will be the frequency of subjects with adequate surgical margins.

Further novel combination therapies with TARE are under investigation. Some evidence suggests that TARE can lead to immune activation in the local tumor microenvironment after treatment [95]. Combination immunotherapy with durvalumab, tremelimumab, and TARE in intrahepatic cholangiocarcinoma is currently under investigation in the United States (NCT06058663) and in Europe (NCT04238637). With the standard of care front-line systemic therapy now including durvalumab or pembrolizumab in addition to gemcitabine–cisplatin in most cases, it will be important to monitor toxicity in combination with TARE based on the high rates of grade 3–4 toxicity seen in the MISPHEC trial.

In summary, TARE outcomes are improving over time with more effective techniques, optimized radiation dosimetry, and refined patient selection. An additional, unique benefit is the ability of the hypertophy liver remnant to enable future liver resection. Limitations include the requirement of a favorable tumor blood supply, which can often only be determined through mapping procedures. Abdominal discomfort and gastrointestinal and pulmonary toxicities have also been seen. Randomized data need to be generated in comparison to systemic therapy and other locoregional techniques to establish its place in the standard of care.

### 1.3. Hepatic Artery Infusion Pump

A hepatic artery infusion pump (HAIP) allows for a continuous flow of intra-arterial chemotherapy to be delivered through a catheter within the gastroduodenal artery, which is a side branch of the hepatic artery. There are two distinct advantages to this approach. Firstly, the hepatic artery is the major source of blood supply for liver tumors, while the portal veins maintain the blood supply of the normal liver [96]. Secondly, floxuridine, a precursor of fluorouracil, can be delivered directly to the liver, where it will subsequently undergo upwards of 95% first-pass metabolism. This allows for up to 400 times higher concentrations of floxuridine to be delivered to the liver compared to the dose received systemically, reducing systemic toxicity and adverse events [97,98]. The pump can be placed in an open or minimally invasive fashion, limiting the time off of systemic therapy and shortening the time to return to treatment after placement.

Most of the early literature surrounding HAIP is derived from a single center, Memorial Sloan Kettering. A 2022 meta-analysis included three phase II trials [99,100,101]. Amongst the 154 unique patients with unresectable intrahepatic cholangiocarcinoam treated with HAIP chemotherapy with floxuridine, the median overall survival was 29.0 months, and 1-, 3-, and 5-years overall survival was 86.4%, 39.5% and 9.7% [102]. Liver-specific PFS has been reported as 11.9 months, underscoring the ability of this therapy to facilitate localized drug delivery [103]. Pump-related complications occur in around 20% of patients historically, with around 10% experiencing pump failure at 1 year [104]. Rates of biliary sclerosis, a feared complication of floxuridine, are generally between 2 and 6% [99,105,106].

Franssen et al. recently published an updated retrospective comparison study. Consecutive patients diagnosed with liver-confined, unresectable (and without previous resection) intrahepatic biopsy-confirmed cholangiocarcinoma were included. Lymph node metastases were allowed. There were 76 patients treated with gemcitabine–cisplatin chemotherapy alone (mainly treated at the MC Cancer Institue in Rotterdam) compared with 192 patients treated with FUDR HAIP with or without systemic therapy (with 71.9% receiving some systemic therapy) at Memorial Sloan Kettering [107]. Overall characteristics were similar between the two groups with notable differences in treatment center, the presence of multifocal liver disease (73.4% in the HAIP group versus 55.3% in the systemic therapy alone group), prior systemic therapy (30.2% in the HAIP group versus 0% in the systemic therapy alone group), and underlying liver disease (5.2% in the HAIP group versus 18.4% in the systemic therapy alone group). The median overall survival for the gemcitabine–cisplatin group was 11.8 months versus 27.7 months for the HAIP group (*p* < 0.0001; HR 0.27 when adjusting for prognostic factors). Three- and five-year overall survival were 3.5% versus 34.3% and 0% versus 15.1% in favor of HAIP. Survival benefits from HAIP seemed similar for those who receive it in first- and second-line settings. In total, 6.8% of patients in the HAIP arm, compared with 1.3% in the systemic therapy arm, received subsequent surgical resection. The 5-year OS was 44.9% for the patients that went for resection. In this non-randomized study, no difference in overall survival was found between those who did and did not receive concurrent systemic therapy with HAIP.

The multi-center phase II PUMP II trial from the Netherlands was presented at the ASCO Gastrointestinal Cancers Symposium in 2024 [108]. A total of 50 patients with unresectable intrahepatic cholangiocarcinoma confined to the liver were planned for treatment with six cycles of FUDR and eight cycles of systemic gemcitabine–cisplatin. The largest tumor diameter on average was 9.2 cm; 66% of patients had multifocal liver disease; 34% had regional lymph nodes; and 22% had prior systemic therapy. The rate of biliary sclerosis requiring stenting was 2% (1 patient), and the post-operative complication rate requiring re-intervention was 22%. The rate of partial response was 46%, with 8% of patients subsequently undergoing resection. One of the patients had a pathological complete response in the resection specimen. The median overall survival was 22 months (compared to 12 months from a historical cohort, *p* < 0.001), and the 3-year overall survival rate was 33% (compared to 3% in the historical cohort). This study provides crucial external validation of the HAIP approach.

In a sample of 83 patients (35 patients from a phase II trial [99] and 48 patients from a routine care cohort) treated with HAIP for intrahepatic cholangiocarcinoma who had tumor sequencing results available, alterations in the TP53 pathway and cell-cycle pathway were associated with worse PFS; whereas, KRAS-TRK and PIK3CA alterations were not [103].

Alternative HAIP chemotherapy regimens have been studied. Cowzer et al. reported responses in 4/9 patients treated with mitomycin C hepatic artery infusion, with a median PFS of 3.93 months [103]. One patient achieved a complete response ongoing at 56 months from the initiation of therapy. Unlike hepatoceullular carcinoma, where a benefit of intrahepatic FOLFOX has been seen [109,110], this regimen did not improve overall survival compared to systemic therapy in a retrospective study of patients with intrahepatic cholangiocarcinoma [111]. Gemcitabine, cisplatin, and 5-fluorouracil have shown some activity in a small cohort of patients [112].

Lin et al. recently published a retrospective study of HAIP chemotherapy with gemcitabine (1000 mg/m^2^ day 1) and oxaliplatin (85 mg/m^2^ day 2) combined with lenvatinib (8–12 mg beginning day 3) and immunotherapy (sintilimab or camrelizumab every 3 weeks beginning day 3) in treatment-naïve patients with advanced intrahepatic cholangiocarcinoma [113]. A cohort of 51 patients, the majority of whom had lymph node metastases (43/51) or Stage IV disease (19/51) undergoing the HAIP protocol were compared to a matched “standard of care” group receiving gemcitabine and cisplatin intravenously. The ORR was 43.1% in the pump group versus 20.5% in the standard of care group, *p* = 0.04. Overall survival was 16.8 months versus 11.0 months in favor of the HAIP group (*p* = 0.01). Progression-free survival was 12.0 months versus 6.9 months in favor of the HAIP group (*p* < 0.01). Treatment was generally well tolerated, with higher rates of ALT/AST elevation (any grade 47% versus 20.6%) and hypertension (35.1% versus 0%) in the HAIP group, though they had fewer total treatment-related adverse events. Although the systemic therapy regimen in this study requires further validation, the fact that the extrahepatic disease subgroup seemed to benefit from HAIP is noteworthy (HR 0.29, 0.13–0.68). It suggests future studies consider the expansion of HAIP to a carefully selected cohort of patients with a low burden of extrahepatic disease.

To our knowledge, there is no direct prospective comparison of HAIP with TARE or other forms of liver-directed therapy. The patient populations differ between the most recent HAIP trials and TARE trials, with bilobar disease often being present in higher proportion in HAIP trials. At least one theoretical advantage to hepatic artery infusional chemotherapy is that it can treat the whole liver, not just sections of visible disease. However, the need for continuous treatment could be seen as a disadvantage.

In summary, hepatic artery infusional therapy remains a promising treatment modality in addition to systemic therapy in unresectable intrahepatic cholangiocarcinoma. A pump program requires multidisciplinary expertise and resources to deliver this complex treatment. We currently lack randomized data showing its benefits. There is an ongoing multicenter randomized trial (NCT04891289) evaluating first-line gemcitabine and oxaliplatin with or without HAIP in patients with unresectable intrahepatic cholangiocarcinoma confined to the liver. Patients with resectable local lymph nodes are eligible for inclusion. Notably, portal vein thrombososis, signs of portal hypertension, Child–Pugh B or greater cirrhosis, sclerosing cholangitis, prior radiation/ablation or systemic therapy are allexclusion criteria for this study. If the results are positive, this trial may shift the treatment paradigm for this patient population. Combinations of sequential HAIP and other local, ablative therapies could also provide disease control benefits in appropriately selected patients.

### 1.4. Transplantation for Intrahepatic Cholangiocarcinoma

Until recently, intrahepatic cholangiocarcinoma remained a contra-indication for transplantation. Initial data suggested very poor outcomes with high recurrence rates and high short-term mortality [114,115]. Renewed interest came from retrospective studies looking at the outcomes of patients who underwent liver transplantation for hepatocellular carcinoma or decompensated cirrhosis who were incidentally found to have intrahepatic cholangiocarcinoma on the explanted liver. Those with “very early” intrahepatic cholangiocarcinoma, defined as a single tumor focus ≤ 2 cm, did relatively well after transplantation. One such multicenter retrospective study (n = 15) reported a recurrence rate of 7%, 18%, and 18% at 1 year, 3 years, and 5 years with a survival rate of 93%, 84%, and 65% at those timepoints [116]. Combining these patients with another cohort, the recurrence rate was estimated to be around 9% for very early intrahepatic cholangiocarcinoma [116,117,118].

Another study evaluated patient outcomes following liver transplantation for hepatocellular carcinoma in those that were found to have intrahepatic cholangiocarcinoma or mixed hepatocellular carcinoma–cholangiocarcinoma on explant pathology [119]. Compared to those with hepatocellular carcinoma (n = 574), those with isolated intrahepatic cholangiocarcinoma (n = 17) or mixed HCC–intrahepatic cholangiocarcinoma (n = 27) had a higher recurrence rate at 36.4% versus 10.8%. Using preoperative imaging findings, those with very early cholangiocarcinoma (≤2 cm, single lesion without vascular invasion) had 1- and 5-year survival rates of 63.6% and 63.6%, with an overall recurrence rate of 33.3%. This was statistically higher than the recurrence rate of patients within Milan Criteria for HCC (11%, *p* =0.02) without a significant difference in 5-year survival (70.3% for the HCC cohort). Of note, preoperative imaging likely understaged intrahepatic cholangiocarcinoma in 66.7% of cases.

Larger tumors (>2 cm) or multifocal intrahepatic cholangiocarcinoma found on the explanted liver have generally been associated with high recurrence risks between 25 and 77% [119,120,121,122,123]. Other characteristics conferring increased risk of recurrence are not uniform between the various studies but include poor differentiation and microvascular invasion [116,121]. A multicenter French retrospective study examined the survival of patients with intrahepatic cholangiocarcinoma or mixed cholangiocarcinoma–HCC with the largest tumor nodules up to 5 cm found on the explant specimen after transplantation for cirrhosis or HCC (n = 49). The patients were compared to a group of patients who underwent resection for intrahepatic cholangiocarcinoma and whose tumors met the same criteria. After a median follow-up of 25 months, the 1-,3- and 5-year survival rates were 90%, 76%, and 67% in the liver transplant group versus 92%, 59%, and 40% in the liver resection group (*p* = 0.17). The rate of recurrence was significantly lower in the transplantation group (27% versus 58%, *p* = 0.008). Contrary to previous studies, they found no difference in recurrence rates between tumors ≤2 cm and >2 but ≤5 cm. Given that intrahepatic cholangiocarcinoma tumors <2 cm that are amenable to surgical resection have a 5-year survival rate of 82% [123], the authors highlight this should likely remain the preferred approach for resectable very early-stage patients. However, the data support the consideration of liver transplantation for intrahepatic cholangiocarcinoma <5 cm that develops in the setting of cirrhosis, or <2 cm and not amenable to resection.

The above retrospective work helped pave the way for prospective trials examining transplantation for intrahepatic cholangiocarcinoma. In 2022, McMillan et al. reported on their study, including 18 patients who underwent liver transplantation for unresectable, locally advanced (>2 cm or multifocal) intrahepatic cholangiocarcinoma [124]. The protocol required 6 months of stability on neoadjuvant therapy prior to transplantation. With a median follow-up of 26 months, the 1-, 3- and 5-year survival for liver transplant patients was 100%, 71%, and 57%, which is significantly improved compared to the patients unable to undergo transplant (*p* = 0.002). The recurrence rate was 38.9% (7/18) in the transplanted patients, with a median time to recurrence of 11 months. At least two cases of recurrence, in retrospect, showed indeterminate pre-transplantation findings that later proved to be metastatic sites of disease. Two recurrences occurred in the setting of R1 resection. The patient population had favorable molecular alterations, including 27% FGFR, 35% IDH1, and 50% with DNA damage repair pathway alterations. Several of these patients were exposed to corresponding targeted therapies. Correlative analysis was limited due to the sample size. A larger cumulative tumor size was associated with an increased risk of death (RR 1.22, *p* = 0.04) among the transplanted patients, and a higher tumor number on preoperative imaging was associated with recurrence during bivariate analysis.

Several additional prospective studies are underway examining transplantation for intrahepatic cholangiocarcinoma (Table 1). Based on the above data, the Organ Procurement and Transplantation Network has approved criteria (Table 1) for liver transplantation for this indication. These criteria are more restrictive than those outlined by McMillan et al. [124], and continued future directions will involve efforts to expand criteria while preserving the outcomes seen in more limited disease. Patient selection in the future may be improved by better imaging modalities and possibly tumor-informed circulating tumor DNA assay, which has recently demonstrated prognostic value in this disease [125]. The optimal sequence and components of neoadjuvant therapy are yet to be defined but will likely involve both locoregional therapy options, as described in previous sections in addition to systemic therapy. As systemic therapy options improve, targeted therapies with superior response rates may be moved to the neoadjuvant setting. Already, since the time of the McMillan publication, immunotherapy has been incorporated as part of front-line therapy. Experience with HCC has shown this to be safe prior to liver transplantation, pending a wash-out period of about 3 months [126].

Recent data from the TRANSMET trial [127] showed a significant improvement in overall survival with liver transplantation for colorectal cancer with unresectable liver metastases, with a 5-year overall survival of 56.6% versus 12.6% in the systemic therapy alone arm. This benefit occurred despite a high rate of recurrence (72.7%, 26/36 patients) in the transplanted group. This study may be important for shifting the paradigm of liver transplantation indication in oncology away from the focus on recurrence rates (or its inverse/cure rate) to the improvement in quality-of-life years instead. It is noted that cholangiocarcinoma, however, is distinct from colorectal cancer, as patients have traditionally done worse following recurrence. The work from recent prospective trials for intrahepatic cholangiocarcinoma and colorectal cancer highlights the role of prolonged stability on neoadjuvant therapy in optimizing patient selection for transplantation. Viable organs for transplantation undoubtedly remain a precious resource. As strategies to improve organ procurement progress, so may the criteria for transplantation for intrahepatic cholangiocarcinoma expand.

## 2. Conclusions

There are now many tools available to treat patients with unresectable intrahepatic cholangiocarcinoma. Locoregional therapies have been shown to be safe in appropriately selected patients. We currently lack randomized clinical trial data demonstrating improvements in survival using various techniques in comparison to systemic therapy alone or in comparison to one another. The recently presented ABC-07 randomized trial notably failed to meet its primary endpoint of PFS improvement for consolidation SBRT after stable disease on front-line systemic therapy, but overall survival data did demonstrate a trend towards improvement, though these results are not yet mature. Outcomes from single-arm prospective studies of local therapy have achieved survival results superior to those typically seen with systemic therapy alone.

There will not be a “one-size-fits-all” approach to locoregional therapy for patients with intrahepatic cholangiocarcinoma. Patient presentation, disease phenotype, and institutional experience will make most treatment determinations and are challenging to study prospectively. Hence, the multidisciplinary tumor board will continue to play a crucial role in selecting the right patients for the right intervention. The art of medicine in these cases is likely to remain beyond inclusion criteria that would allow for the direct head-to-head comparison of techniques for all patients. Table 2 highlights some recently reported seminal prospective trials for different locoregional therapy options and clinical scenarios where such treatments may be pursued or avoided.

Generally, locoregional therapy has the best supportive evidence for patients with unresectable liver-limited disease when combined with systemic therapy in an upfront setting. It may also be appropriate to consider these treatments in cases with progressive disease in the liver with limited remaining systemic therapy options or those not fit for systemic therapy.

The ultimate goal is to improve survival for this aggressive disease. Transplantation seems to show the best chance of long-term survival in carefully selected patients with disease that is unresectable using traditional surgical approaches. Treatment with a combination of systemic and locoregional therapies followed by a period of stable disease may allow for optimal selection and the achievement of best results.

## 3. Future Directions

Further randomized data are highly anticipated for locoregional therapies in unresectable intrahepatic cholangiocarcinoma. Improvements in the precise delivery of local therapy, such as proton and carbon ion radiation and selective ablative TARE, may open up treatment options for more patients. Alternative methods for the targeted delivery of radiopharmaceuticals, such as theranostic, may also prove important [128,129]. The refinement of imaging and molecular data, possibly through new artificial intelligence tools and biomarkers such as ctDNA, will further improve patient selection. Combination strategies with newer systemic therapy options such as immune-checkpoint inhibitors, targeted therapies and locoregional therapies may lead to further incremental gains in survival for unresectable cholangiocarcinoma.

## Figures and Tables

**Table 1 curroncol-32-00082-t001:** Proposed criteria for the transplantation of intrahepatic cholangiocarcinoma from the Organ Procurement and Transplantation Network compared to reported or currently recruiting prospective trials.

	OPTN Board	Methodist-MD Anderson Protocol	TESLA Trial (NCT04556214)	NCT04195503	NCT06140134	**LIRICA Trial (NCT06098547)**
Inclusion criteria	Biopsy-proven intrahepatic cholangiocarcinoma Unresectable size ≤3 cm Six months of therapy with stable disease before initial exception request	Biopsy-proven intrahepatic cholangiocarcinoma Locally advanced tumors: ≥2 cm or multiple tumors Unresectable due to tumor extent or underlying liver disease (including after neoadjuvant treatment) No evidence of extrahepatic disease, lymph node involvement, or encasement/involvement of major vascular structures (assessed with PET-CT, MRI, bone scan, and EUS-guided biopsy of enlarged nodes) Prior resection allowed if >6 months from listing Stable disease for ≥6 months on given regimen	Biopsy-proven intrahepatic cholangiocarcinoma Unresectable based on tumor location or underlying liver dysfunction No evidence of extrahepatic disease, lymph node involvement, or vascular invasion detected on imaging (assessment including PET-CT) Twelve months or more from diagnosis Prior resection allowed No progressive disease at listing	Biopsy-proven intrahepatic cholangiocarcinoma Unresectable disease based on tumor location or underlying liver dysfunction No evidence of extrahepatic disease, lymph node involvement, or vascular invasion Stable disease for ≥6 months	Biopsy-proven intrahepatic cholangiocarcinoma Unresectable disease based on tumor location or underlying liver disease No evidence of extrahepatic disease, lymph node involvement, or vascular invasion Stable disease for ≥6 months on current therapy (if second-line therapy must also have ≥6 months of disease control on that regimen)	Biopsy-proven intrahepatic cholangiocarcinoma Unresectable disease based on tumor location, extent, or underlying liver dysfunction No evidence of extrahepatic disease, involvement of extrahepatic structures, including visceral peritoneum, or major hepatic vessels assessed by PET-MR and CT At least 6 months from diagnosis or recurrence (with original R0, N0 resection) Stable disease ≥ 6 months of systemic therapy
Neoadjuvant treatment	Unspecified	First-line platinum-based therapy + gemcitabine +/− biologic or second-line therapy if progressive or intolerant Locoregional therapy if recommended from MDT	Received at least 6 months of chemotherapy or locoregional therapy	≥6 months of gemcitabine-based therapy Locoregional therapy is not permitted	≥6 months of neoadjuvant therapy 2nd-line therapy is allowed	≥6 months of neoadjuvant therapy
Adjuvant treatment	Unspecified	4–6 months of adjuvant treatment if viable tumor on explant (and based on MDT discussion)	Unspecified	Unspecified	Unspecified	Unspecified
Transplant type	Unspecified	Mainly extended criteria donors (deceased donors or domino living donors) Staging Laparoscopy performed at the time of transplantation	Unspecified	Living donor	Whole-liver allotransplantation	Cadaveric or living donor, whole or partial liver
Outcomes		N = 14 5-year survival in transplanted patients: 57% Recurrence rate after transplantation: 38.9%	Enrollment goal: N = 15 Primary endpoint: overall survival	Estimated enrollment of 10 participants Primary endpoint: 5-year overall survival	Target accrual: unknown Primary endpoint: 5-year overall survival	Target accrual: unknown Primary endpoints: 3- and 5-year overall survival

MDT = multidisciplinary team; OPTN = Organ Procurement and Transplantation Network; EUS—endoscopic ultrasound; PET-CT = positron-emission tomography—computed tomography; MRI = magnetic resonance imaging.

**Table 2 curroncol-32-00082-t002:** Highlighted recent prospective publications, clinical scenarios for optimal use, and avoidance of various locoregional therapies as part of a multidisciplinary approach to management.

		SBRT	TARE	HAIP
Highlighted Recent Prospective Multicenter Trials		ABC-07 trial	MISPHEC trial	PUMP II trial
	Inclusion Criteria	Inoperable, liver-limited, and no progression after 3 months of gemcitabine–cisplatin	Inoperable disease, CP ≤ 7, hilar lymph node ≤3 cm or <5 lung nodules (all ≤10 mm), with no previous chemotherapy	Inoperable disease, confirned to the liver (regional lymph nodes allowed)
	Intervention	2:1 randomization of SBRT after 6th cycle vs. continued chemotherapy up to 8 cycles	Single-arm study: Y90 and gemcitabine–cisplatin	Single-arm study: 8 cycles of gemcitabine–cisplatin with HAIP FUDR for 6 cycles
	Sample Size	N = 45	N = 41	N = 50
	Outcomes	PFS: 8.6 m with SBRT vs. 9.0 m (HR = 1.0, *p* = 0.99) OS: 19.4 m with SBRT vs. 14.2 m without (HR = 0.79, *p* = 0.47)	ORR: 39% PFS: 14 m (8–17) OS: 22 m (14–52).	ORR: 48% PFS: 10.0 m (8.7–12.2) OS: 22.1 m (19.7-NR)
Our Favored Clinical Scenario for Use		A unilobar or single-segment disease where ablative dosing can be achieved; significant local lymph node burden; this is generally preferred over alternative locoregional therapies in the setting of poor liver function due to its safety profile *	Segmental, divisional, lobar, and trisegmetnal disease Generate liver hypertrophy in the setting of inadequate FLR to enable resection; this is preferred as an ablative modality, particularly near critical structures	Multifocal and bilobar disease
Limitations		Large tumor volumes if underlying liver disease; tumor locations close to crucial structures (bowel and heart)	Poor vascular mapping; avoid pairing with chemotherapy if underlying cirrhosis	Avoid in PSC or portal vein thrombosis

* Proton beam therapy is an acceptable alternative to SBRT where it is available and may be preferred in certain situations where there is a higher risk of damage to surrounding tissue. SBRT = stereotactic body radiation therapy; TARE = transarterial radioembolization; FLR = future liver remnant; HAIP = hepatic artery infusion pump; m = months; ORR = objective response rate; PFS = progression-free survival; OS = overall survival; FUDR = floxuridine; NR = not reached; PSC = primary sclerosing cholangitis.

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
