# Peer review of "Advancements in Locoregional Therapies for Unresectable Intrahepatic Cholangiocarcinoma"

_curroncol, 2025, doi:10.3390/curroncol32020082_

Round 1

Reviewer 1 Report

Comments and Suggestions for Authors

Thanks for the interesting review. See my comments below.

1. Authors have outlined nicely the advantages of SIRT. If authors can state some of the limitations with SIRT that will be great. Example SIRT can potentially damage lungs due to hepatopulmonary shunting.

2. Not every patient can undergo SIRT, hence, careful selection of patients and pre-treatment are critical to mitigate risk associated with SIRT.

3. Do authors know what will happen to a patient if the glass microsphere reaches the stomach.

4. Aside the 90Y glass microsphere is there any radiopharmaceutical agent that has been used to treat intrahepatic cholangiocarcinoma.

5. Paragraph 259: Please, change Y90 to 90Y. Do this for all Yttrium-90 abbreviations.

6. Paragraph 47-48: Why is the reoccurrence rate for intrahepatic cholangiocarcinoma treatment high in this case.

7. Radiopharmaceuticals are revolutionizing cancer space at this moment. It may be a good idea if authors could include this in your future directions. 

8. Authors can look at this interesting review talking about radiopharmaceuticals and reference it if possible; https://www.mdpi.com/1999-4923/13/5/599.

Author Response

Comment #1: Authors have outlined nicely the advantages of SIRT. If authors can state some of the limitations with SIRT that will be great. Example SIRT can potentially damage lungs due to hepatopulmonary shunting.

Response #1: This is a good point. We have added a sentence in the summary for this section (Line 410):” Limitations include the requirement of favorable tumor blood supply, which can often only be determined through mapping procedures. Abdominal discomfort, gastrointestinal and pulmonary toxicities can be seen.”

Comment #2: Not every patient can undergo SIRT, hence, careful selection of patients and pre-treatment are critical to mitigate risk associated with SIRT.

Response #2: Agree with this point and hope the alteration above stresses this point further.

Comment #3: Do authors know what will happen to a patient if the glass microsphere reaches the stomach.

Response #3: There is some data on this. We have added sentence and reference to address this (line 223): “Non-target delivery of miscrospheres to the gastrointestinal tract can lead to injury to the gastric mucosa and radiation-induced ulceration and bleeding [71]”.

Comment #4: Aside the 90Y glass microsphere is there any radiopharmaceutical agent that has been used to treat intrahepatic cholangiocarcinoma?

Response #4: Yes, other agents have been used. We have added the following sentence and reference (Line 234): “Alternative therapeutic particles such as Y90-resin and Ho-labeled poly(l-lactic acid) microspheres have also been studied[70]."

Comment #5: Paragraph 259: Please, change Y90 to 90Y. Do this for all Yttrium-90 abbreviations.

Response #5:Thank you for the observation. I found one case of Yttrium-90 being written after it was abbreviated in the text and changed this to abbreviation (Line 398). The example from paragraph around 259 was alluding to the title of a trial and so I have placed this in quotations. In an effort to be consistent across our previous publications, we are asking to keep the abbreviation as Y90 rather than 90Y.

However, given your well taken point about other forms of radiopharmaceuticals being available/under study which deliver selective radiation, we have changed the term in our paper from SIRT to transarterial radioembolization (TARE) to more specifically reflect the intervention in question.

Comment #6: Paragraph 47-48: Why is the reoccurrence rate for intrahepatic cholangiocarcinoma treatment high in this case.

 This is likely driven in part by tumor biology factors that are poorly understood. We have added information on additional aspects that make surgery difficult, which may  contribute to poor outcomes.

Added Line 66: “An additional challenge given the typical locally advanced presentation of intrahepatic cholangiocarcinoma is the requirement of adequate future liver remnant in patients after resection to prevent post-hepatectomy liver failure [21]. Even with R0 resections with adequate liver remnant, poor tumor biology remains a competing risk for early disease recurrence.”

Comment #7: Radiopharmaceuticals are revolutionizing cancer space at this moment. It may be a good idea if authors could include this in your future directions. 

Response #7:  Great idea. Thank you for the suggestion. Line added (659): "Alternative methods for targeted delivery of radiopharmaceuticals such as theranostic may also prove important [129,130]."

Comment #8: Authors can look at this interesting review talking about radiopharmaceuticals and reference it if possible; https://www.mdpi.com/1999-4923/13/5/599.

Response #8: This has been added to the text in the response to comment 7.

Please allow for consideration a few additional modifications we are suggesting:

  • Paragraph beginning on 231: Details of mapping angiography were adjusted to be more technically correct.
  • Line 262: An important prospective trial highlighting the different outcomes between transarterial radioembolization for solitary lesion and multifocal lesions was added.
  • Paragraph beginning on line 352 describing a retrospective study was removed for redundancy and to try to keep the sections closer to equal in their coverage.
  • Line 380: The concept of radiation lobectomy was discussed in more detail as this is thought to be a critical application of transarterial radioembolization not sufficiently covered in the original version.
  • Table 2: Favored clinical scenarios for transarterial radioembolization were updated to include more technical terminology.
  • A seminal paper was just published questioning the benefit of adjuvant therapy in intrahepatic cholangiocarcinoma. Line 55 adds this reference.

Reviewer 2 Report

Comments and Suggestions for Authors

This is a well-written review of the current evidence for four treatment modalities for IHC: EBRT, SIRT, HAIP chemo, and liver transplant. The summaries of studies for each modality in a separate section is excellent. The EBRT section is shorter than the others; one relevant piece of information that could be added is more guidance on liver and normal tissue dose constraints for SBRT, as that can help determine when SBRT is safe. I suggest the authors add a reference to the ASTRO liver guidelines paper, which includes IHC: https://doi.org/10.1016/j.prro.2021.09.004.

Table 2 is a nice summary for practitioners to reference when considering management options. I expected to see proton beam therapy in the table, but I suppose that was left out because there's no clinical trial evidence.

Thanks for the nice paper.

Author Response

Comment 1: " This is a well-written review of the current evidence for four treatment modalities for IHC: EBRT, SIRT, HAIP chemo, and liver transplant. The summaries of studies for each modality in a separate section is excellent. The EBRT section is shorter than the others; one relevant piece of information that could be added is more guidance on liver and normal tissue dose constraints for SBRT, as that can help determine when SBRT is safe. I suggest the authors add a reference to the ASTRO liver guidelines paper, which includes IHC: https://doi.org/10.1016/j.prro.2021.09.004."

Response 1: 

-Thank you for the constructive feedback. We have added some information from this reference. Line 168: “The American Society for Radiation Oncology (ASTRO) clinical practice guidelines suggest dose constraints to uninvolved liver and bowel structures to maintain safety [61]. For example, a mean dose < 15-18Gy is recommended to the uninvolved liver in non-cirrhotic patients receiving 5 fraction treatments compared to <13-15Gy for those with Child Pugh A cirrhosis.  Maintaining these dose constraints limits the risk of radiation induced liver disease. Dosing to the stomach and duodenum should be kept below 32Gy in a 5 fraction delivery to minimize the risk of luminal organ ulceration [61]. “

Comment 2: "Table 2 is a nice summary for practitioners to reference when considering management options. I expected to see proton beam therapy in the table, but I suppose that was left out because there's no clinical trial evidence."

Response 2: 

-Great point. The reason it wasn’t included originally was to avoid confusion with regard to the trial summary aspect of the table.

-An asterisk has been added to the Table 2:

“*Proton beam therapy is an acceptable alternative to SBRT where available and may be preferred in certain situations where there is higher risk of damage to surrounding tissues.” 

Please allow for consideration a few additional modifications we are suggesting:

  • Paragraph beginning on 231: Details of mapping angiography were adjusted to be more technically correct.
  • Line 262: An important prospective trial highlighting the different outcomes between transarterial radioembolization for solitary lesion and multifocal lesions was added.
  • Paragraph beginning on line 352 describing a retrospective study was removed for redundancy and to try to keep the sections closer to equal in their coverage.
  • Line 380: The concept of radiation lobectomy was discussed in more detail as this is thought to be a critical application of transarterial radioembolization not sufficiently covered in the original version.
  • Table 2: Favored clinical scenarios for transarterial radioembolization were updated to include more technical terminology.
  • A seminal paper was just published questioning the benefit of adjuvant therapy in intrahepatic cholangiocarcinoma. Line 55 adds this reference.